# Evaluation of Non-Fermented and Fermented Chinese Chive Juice as an Alternative to Antibiotic Growth Promoters of Broilers

**DOI:** 10.3390/ani12202742

**Published:** 2022-10-12

**Authors:** Woo-Do Lee, Damini Kothari, Seung-Gyu Moon, Jongil Kim, Kyung-Il Kim, Gun-Whi Ga, Yong-Gi Kim, Soo-Ki Kim

**Affiliations:** 1Department of Animal Science and Technology, Konkuk University, Seoul 05029, Korea; 2Poultry Research Institute, National Institute of Animal Science, RDA, Pyeongchang 25342, Korea

**Keywords:** broiler, *Lactobacillus plantarum*, fermented *Allium tuberosum*, phytogenic feed additive, alternative antibiotic

## Abstract

**Simple Summary:**

Rampant use of growth-promoting antibiotics in the poultry industry has resulted in enhanced harmful concentrations in livestock products ushering in the evolution of multidrug resistance strains. Plant feed additives are being explored, and among them, Chinese chives (CC), which contain various phytonutrients, can offer a promising alternative. We performed a comparative study to examine the effects of non-fermented and fermented CC feed additives on broiler productivity, chicken meat quality, blood composition, intestinal properties, and gut microbiota. We observed a high broiler productivity for the treatment group subjected to fermented CC juice on par with growth-promoting antibiotics. Fermented CC juice influenced low cholesterol content in the broilers while inhibiting the growth of intestinal pathogens.

**Abstract:**

The present study explores the application of CC juice as a suitable feed additive and alternative to conventional antibiotics. We performed a comparative study to investigate the effects of non-fermented and fermented CC juice on broiler productivity, meat quality, blood characteristics, intestinal characteristics, and microbiota associated with intestinal characteristics. A total of 800 one-day-old Ross 308 broiler chickens were randomly assigned to one of the four dietary treatment groups: (1) basal diet (negative control; NC); (2) basal diet + 0.01% enramycin (positive control; PC); (3) basal diet + 3% non-fermented CC juice (NCC; CC juice 10%, water 90%); and (4) basal diet + 3% fermented CC juice (FCC; CC juice 10%, water 90%, *Lactobacillus plantarum* SK4719). Feed and water were provided *ad libitum*. Intriguingly, all treatments showed similar results in terms of broiler productivity and chicken meat quality. Considering organ characteristics, the FCC group showed a low spleen weight and lower (*p* < 0.05) blood levels of AST and total cholesterol (TCHO). Regarding intestinal characteristics, the CC feed additive (NCC and FCC) resulted in a heavier intestinal weight (*p* < 0.05) without affecting the length ratio of the villi or the crypt compared to the control (NC or PC). NCC and FCC lowered the growth of intestinal pathogens (*p* < 0.01). In summary, the addition of FCC can maintain poultry health by improving blood compositions and inhibiting the growth of intestinal pathogens, leading to a productivity comparable to that of poultry treated with growth-promoting antibiotics.

## 1. Introduction

Antibiotic-based growth promoters have been used in animal feeds for a long time to promote growth and improve digestive capacity [1,2]. Representative antibiotics commonly used in animal feed include virginiamycin, salinomycin, enramycin, avilamycin, monensin, and narasin [3,4]. In the poultry industry, the use of antibiotics can improve health and well-being by promoting efficient production and reducing disease outbreaks. Unfortunately, eggs and chicken meat can be contaminated by harmful concentrations of drug residues [5]. Accordingly, continuous research is being conducted to find an antibiotic substitute that has antimicrobial and growth-promoting effects that can simultaneously avoid the hazards of antibiotic resistance in livestock and their end consumers. Alternatives include organic acids, enzymes, probiotics, prebiotics, and phytogenic compounds [6]. Phytogenic compounds are defined as natural bioactive compounds derived from plants (essential oils & phytoextracts) with positive effects on animal growth and health [7,8]. Some botanical compounds are known to have antibacterial, antiviral, antifungal, and antioxidant properties. They have been traditionally used as complementary or alternative medicines to improve health or treat diseases [9,10].

*Allium* species such as Chinese chives (CC), garlic, onions, shallots, and rakkyo have been used as foods, spices, and herbal remedies across various civilizations throughout the world [11,12]. The genus *Allium* has more than 600 species distributed all over Europe, North America, Northern-Africa, and Asia [13]. *Allium* species are known for their different taste, aroma, color, and visual appearance. They are rich sources of bioactive compounds including organosulfur compounds, polyphenols, saponins, fructans, and fructo-oligosaccharides [13,14]. The CC (*Allium tuberosum*) is a perennial plant cultivated in many countries in Asia [15]. Recent literature indicates that CCs have a broad spectrum of antimicrobial activities against a range of pathogenic bacteria and fungi. They also possess insecticidal activity [16,17]. Various species of CCs have high contents of polyphenols, antioxidant vitamins, and organosulfur compounds with antibacterial, anticholesterol, anticoagulation, and anticancer activities [18,19]. Fermentation can improve the nutritional quality of substrates by increasing the bioavailability of phytonutrients coupled with the biotransformation of anti-nutritional factors in the feedstuff [20,21]. Fermentation can preserve food from spoilage. It also helps improve the flavor, digestibility, nutritional content, and palatability of substrates [22]. In particular, controlled fermentation can significantly improve desirable biochemical characteristics including the antioxidant and antipathogenic activities of food [23]. Lactic acid bacteria (LAB) belonging to the genus *Bacillus*, yeasts, and fungi are commonly used for food fermentation [24,25,26,27]. Among them, LABs are the most accepted and safe microbial species used in commercial fermentation bioprocesses for foods, including milk, bread, vegetables, and meat [28,29]. In addition, LABs, including *Lactobacilli*, *Enterococci*, and *Leuconostoc* have biological functions such as antibacterial, anticancer, antidiabetic, antiobesity, and antioxidant activities [30,31]. As a result of analyzing changes in physiologically active substances according to the fermentation of CC juice through ultra-high-performance liquid chromatography-linear trap quadrupole-orbitrap-tandem mass spectrometry (UHPLC-LTQ-Orbitrap-MS/MS) analysis in a previous study, functional substances such as propanoic acid, benzoylmesaconine, kaempferol, isorhamnetin, quercetin, hydroxystearic acid, and saponins were improved [17]. These substances have been reported to have many functions such as analgesic, antinociceptive, antiviral, antifungal, antioxidant, antiradical, inducing apoptosis, etc [32,33,34,35,36,37,38,39,40]. In fact, CC juice fermentation through *Lactobacillus plantarum* SK4719 improved the ability to inhibit avian influenza virus and poultry pathogens [17].

Although CCs are known to possess various bioactivities, effects of fermentation on the bioactivities of CCs on the growth of poultry remain unclear. Therefore, the aim of the present study is to investigate effects of CC juice supplementation (fermented using *L*. *plantarum* SK4719 and non-fermented) on the growth performance, meat quality, blood profiles, immune indicators, and gastrointestinal characteristics of broiler chickens. Another aim of this study is to reveal the antibiotic substitution effects of phytogenic feed additives by designing an antibiotic treatment group for promoting the growth of poultry.

## 2. Materials and Methods

### 2.1. Ethical Statement

The experimental protocol was reviewed and approved by the Institutional Animal Care and Use Committee (Approval number: KU21178) of Konkuk University (Seoul, Republic of Korea). 

### 2.2. Feed Additives and Feed Composition

The CCs used in the experiment were purchased from a local market (Gwangjin-gu, Seoul). CC juice was extracted using a juicer (Angel-7700, Busan, Republic of Korea). The heat treatment, concentration, and fermentation strain of the CC juice were determined by referring to previous studies [17,41]. Diluted juice (CC juice 10% and water 90%) was used for making non-fermented and fermented feed additives. Fermentation was carried out at 30 °C for 18 h with shaking at 100 rpm using *L*. *plantarum* SK4719 as an inoculum (number of *L*. *plantarum* SK4719 in final product: over than 10^8^ colony forming units (CFU)/mL). Enramycin, a Woogene B&G (Hwaseong-si, Gyeonggi-do, Republic of Korea) product, was used as an antibiotic.

The feeding experiment was divided into two stages: (1) feeding the starter feed from day 1 to day 21 and (2) feeding the grower feed from day 22 to day 35. Ingredients and calculated chemical compositions of the corn and soybean meal-based basic feeds are shown in Table 1, formulated according to the 2017 Korean poultry feeding standard [42].

### 2.3. Experimental Design

The following experimental groups were designed: (1) broiler chickens fed a basal diet (negative control; NC); (2) broiler chickens fed a basal diet supplemented with enramycin (positive control; PC); (3) broiler chickens fed a basal diet supplemented with non-fermented CC juice (NCC; CC juice 10%, water 90%); and (4) broiler chickens fed a basal diet supplemented with fermented CC juice (FCC; CC juice 10%, water 90%, *L*. *plantarum* SK4719). NCC and FCC were added at a level of 3% of the basal diet, referring to the effective additive concentration in other *Allium* plants (garlic, hooker chives, and onions) in the reported study [43,44,45,46,47]. Enramycin was added at 0.01% of the broiler feed. Each feed additive was mixed with the feed using a feed mixer (HC 123132-500L, Korea) for 10 min.

### 2.4. Bird Management

In this study, a total of 800 one-day-old male Ross 308 broiler chicks were purchased from a commercial hatchery (Chunan-si, Gyeonggi-do, Korea). The experimental period was conducted for a total of 5 weeks from August to September 2019. All birds were given an adaptation period of one week, and the experiment was performed for a total of 4 weeks from 8 to 35 days old. All birds were kept in a pen in which the floor was covered with rice hulls 5 cm thick for 5 weeks, including a one-week adaptation period. During the adaptation period, all chicks were given the same starter diet, and after the adaptation period, they were randomly assigned to 4 treatment groups (five replicates for each treatment, 40 birds per replicate having a similar body weight (BW): 134.5 ± 0.2 g). The temperature in the house was set to 32 ± 1 °C for the first week and then decreased by 1~2 °C every week to maintain 25 ± 1 °C at the end of the experiment. The relative humidity was set to 60~65%. In addition, ventilation fans were operated and maintained to meet the recommended temperature and humidity. The light was turned on for 24 h continuously from the start of the experiment to the end of the experiment. Feed in mash form and water were provided ad libitum.

### 2.5. Growth Performance

To investigate the productivity of the broilers, the average daily feed intake (ADFI), the average daily weight gain (ADG), and the feed:gain (F:G) ratio were measured. BW and feed intake were measured on a weekly basis from the start of the experiment to the end of the experiment. BW was calculated as the average body weight per bird by measuring all the stocking birds in one pen. The ADFI was measured once weekly per replicate by weighing the amount of feed dispensed and the amount of residual and scattered feed. The F:G was calculated as the ratio of the ADG to the ADFI for each week.

### 2.6. Sample Collection

At the end of the feeding experiment, samples were collected by randomly selecting two birds per replicate. The samples collected were blood, breast, thigh, liver, spleen, bursa of Fabricius, small intestine (duodenum, jejunum, and ileum), and ceca. Blood was collected by cardiac puncture after CO_2_ euthanasia into an EDTA-treated blood collection tube and stored at 4 °C. The plasma was separated by centrifugation at 1500 rpm for 10 min and stored at −20 °C until analysis. Breast, thigh, liver, spleen, bursa of Fabricius, small intestine, and ceca samples were stored at 4 °C immediately after slaughtering.

### 2.7. Meat Quality

The relative weight of the chicken meat was expressed as a ratio of the weight of the chicken meat per 100 g of live weight by measuring the weight of the broilers selected from each treatment group and the weight of the collected breast and thigh meat of the broilers. Meat color was analyzed using a colorimeter (chroma meter CR-210, Konica Minolta Inc., Tokyo, Japan) on the surface of each sample. Among color items, L* means lightness, a* means redness, and b* means yellowness. The standard color was a calibration plate with an L* value of 97.69, an a* value of −0.43, and a b* value of 1.98. The pH was measured as the value displayed when a pH meter (Hanna Instruments, Nusfalau, Romania) was inserted at a depth of 1 cm into the breast or thigh. It was measured three times per sample. For cooking loss, after shaping the sample into a circular and constant shape, it was placed in a polyethylene bag and heated to 75 °C in a water bath (C-WBE, Changshin Science Co., Seoul, Republic of Korea) for 30 min. It was then allowed to cool at room temperature for 10 min. The weight loss was calculated by comparing the difference in weight before and after heating.

### 2.8. Organ Weights

The weights of dissected organs (liver, spleen, and bursa of Fabricius) were measured using an electronic balance (EL4002, Mettler Toledo, Zürich, Switzerland) after removing fat and were expressed as the ratio of weight per 100 g of live weight.

### 2.9. Blood Profiles

The chemical composition of the blood was analyzed using an automated dry chemistry analyzer for veterinary use (CHEM 7000i, Tokyo, Japan) located at the Konkuk University Bio Center for Research Facilities (Gwangjin-gu, Seoul, Korea). Analysis parameters were aspartate aminotransferase (AST), alanine aminotransferase (ALT), gamma(γ)-glutamyl transferase (GGT), alkaline phosphatase (ALP), blood urea nitrogen (BUN), creatinine (CRE), uric acid (UA), lactate dehydrogenase (LDH), total cholesterol (TCHO), triglycerides (TG), high density lipoprotein-cholesterol (HDL-C) (mg/dL), HDL-C (% total), and low-density lipoprotein-cholesterol (LDL-C) + very-low-density lipoprotein-cholesterol (VLDL-C) [48].

### 2.10. Investigation of Intestinal Characteristics

The small intestine and ceca were used to analyze intestinal characteristics. The small intestine was divided into three parts: the duodenum, jejunum, and ileum. The segments of the small intestine were defined as follows: (1) duodenum: from the gizzard to the beginning of the mesentery, (2) jejunum: from the most distal point of insertion of the mesentery to 5 cm before Meckel’s diverticulum, and (3) ileum: from 5 cm after Meckel’s diverticulum to the ileocecal junction [49]. The duodenal, jejunal, ileal, and cecum segments were stored at 4 °C. The lengths and weights of the duodenum, jejunum, ileum, and cecum were recorded. Their weights were expressed as percentages of live weights. After removing the intestinal contents, the middle complete jejunal and ileal segments with lengths of 3 cm were collected and fixed in 10% buffered formalin for at least 48 h for further histological processing. Sections were stained using standard hematoxylin–eosin (H&E) staining. Villus height (VH) and crypt depth (CD) were determined using a microscope (BX43, Olympus, Tokyo Japan) and software (eXcope software, Seoul, Republic of Korea). The VH was measured from the villi vertex to the villi bottom. The CD was defined from the bottom of the villi to the crypt. Their ratio (VH:CD) was calculated by taking the VH and the CD into account [50,51].

### 2.11. Analysis of Microbial Flora in the Small Intestine and Ceca

The number of viable bacteria in the small intestine and ceca was measured with the standard agar plating method using de Man, Rogosa, and Sharpe (MRS), MacConkey, *Streptococcus thermo**philus* (ST), *Salmonella shigella* (SS), and nutrient agar (NA) (Difco, Franklin Lakes, NJ, USA). Contents of the intestine used ten birds per treatment. After 1 g of each sample was added to 9 mL of sterilized distilled water, it was homogenized and sequentially diluted. After that, 10 uL were dispensed into each medium, spotted, and cultured in an incubator at 37 °C for 24 h. By counting the number of effective colonies shown, the CFU/g was calculated and converted into a log_10_ value. All experiments were carried out in triplicates.

### 2.12. Statistical Analysis

All data were subjected to analysis of variance (ANOVA) using the general linear model (GLM) function of an SAS 9.4 (SAS Institute, Cary, NC, USA). The replicates (40 chicks each) were the experimental units for analysis of the performance parameters (BW, ADG, ADFI, and F:G). Individual birds served as the experimental units for meat quality, blood parameters, organ characteristics, and intestinal microbiota. Duncan’s multiple range test was carried out to assess significant differences for all the analyzed parameters at the probability level of *p* < 0.05 among experimental groups. Data are presented as least square means and standard errors of means (SEM).

## 3. Results

### 3.1. Growth Performance

The growth performance parameters of the broilers according to dietary treatments are shown in Table 2. The BW, ADG, and ADFI showed marginal differences among the different treatment groups. For the F:G, the PC group showed lower values during the starter period (*p* < 0.05). However, over the whole period, all treatments showed similar productivity traits of broilers.

### 3.2. Meat Quality

Table 3 presents the effects of dietary CC juice supplementation on chicken meat quality of broilers. There were no differences in the production of breast or thigh meat or heat loss among the treatment groups. There were no significant differences in the color values of the chicken meat samples among all the treatment groups except for the b* value of thigh meat. The pH of the chicken was not affected by dietary treatment either.

### 3.3. Organ Weight

The relative weights of the liver, spleen, and bursa of Fabricius are presented in Table 4. The weights of the liver and bursa of Fabricius did not show any significant differences among the treatment groups. However, the positive control group and the FCC group showed lower spleen weights than the NC group (*p* < 0.05).

### 3.4. Blood Parameters

The changes in blood biochemical parameters for broilers in the different treatment groups are summarized in Table 5. As a result of analyzing items indicating the presence or absence of diseases, such as hepatocellular disease or hepatitis, the PC group showed higher AST levels (*p* < 0.05). Blood TCHO levels (*p* < 0.001) and HDL-C content (*p* < 0.05) were lower in the FCC group than in other groups. However, there was no significant difference in the proportion of HDL-C (% total) among the treatment groups. Uric acid was higher in the NCC group (*p* < 0.01).

### 3.5. Intestinal Parameters

As a result of analyzing the lengths of the intestines, the relative length of the jejunum was shorter in the PC group (*p* < 0.05). There were no significant differences in the relative lengths of the duodenum, ileum, or ceca (Table 6). The relative weights of duodenum, jejunum, and ileum were heavier in the CC groups (NCC and FCC) than in the PC group (*p* < 0.05).

### 3.6. Histological Analysis

The results of the histological analysis of the intestines (jejunum and ileum) for each treatment group are shown in Table 7. As a result of comparing the lengths and ratios of the villi and the crypts of the jejunum, there were no significant differences among the treatment groups. In the case of the ileum, the NC group and the FCC group showed higher crypt depths (*p* < 0.05). However, the ratio of the villi to the crypts was similar among all the treatment groups.

### 3.7. Comparison of the Total Number of Intestinal Bacteria

Table 8 shows the changes in the microflora in different intestinal sections of the broilers according to the dietary treatments. As a result of the analysis, MacConkey and SS (selective media for pathogens such as *E*. *coli*) showed lower numbers of bacterial counts in the FCC group, whereas the PC group showed higher numbers in the jejunum and ileum (*p* < 0.01). In the MRS medium, in which lactic acid bacteria were grown, the FCC group showed a high bacterial population (*p* < 0.05).

## 4. Discussion

As the use of antibiotics is banned, the development of plant feed additives that can replace antibiotics in the poultry industry is required [17]. Among plants being studied recently, plants of the genus *Allium*, including garlic, onions, shallots, ramps, and CCs, contain various bioactive compounds such as polyphenols, flavonoids, organosulfur compounds, saponins, and fructans [52]. Puvača et al. [46] reported that sulfur compounds (e.g., allyl methyl thiosulphonate, 1-propenyl allyl thiosulphonate) present in plants of the genus *Allium* can affect the feed intake, feed efficiency, and body weight in broilers. Rehman and Munir [53] reported that antibacterial, anti-inflammatory, and antiseptic effects of plants such as garlic can have a very positive effect on the growth of broilers and improve feed efficiency by developing intestinal villi. Goodarzi et al. [44] also reported that when onion bulbs are used as a feed additive, feed intake and final weight are significantly higher than those of the control and virginiamycin groups. The addition of *A*. *hookeri* leaves (0.3% and 0.5%) for 35 days can also significantly increase the body weights of broilers [54]. The addition of 2% garlic powder for 42 days resulted in a higher final body weight than the control (control: 1949.5 g; 2% garlic powder: 2019.0 g) [55]. However, some studies have shown comparable broiler productivity despite the addition of various types (e.g., garlic powder, garlic extract, chive extract, and onion peel aqueous extract) of plants of the genus *Allium* [56,57,58,59,60,61]. In the results of this study, when the CC additive was mixed with feed, it showed no significant difference in productivity with enramycin, a poultry growth promoting antibiotic. Based on these results, it is considered that the CC additive has the potential to replace growth-promoting antibiotics. 

Recently, as mass production of broilers has been achieved, research is being conducted on how to improve the quality of the broilers [62]. The quality of chicken meat, such as taste, smell, color, and appearance, is a characteristic that affects consumers’ purchasing decisions and preferences and is also affected by the type of feed additives in the feed [62,63]. Among them, there was a report that physiologically active compounds such as polyphenols contained in plant extracts and herbs act as antioxidants which improve the quality of the chicken meat [63]. In recent years, research on the effect on quality by adding plants of the genus *Allium* is being conducted. Kirkpinar et al. [64] significantly reduced the L* value of chicken meat when oregano and garlic oil were added to the broiler feed. In another study, the addition of onion extract had a significant effect on the color (L*, a*, and b*) of chicken meat [65]. Ao et al. [66] reported that when garlic powder fermented with *Weissella koreensis* was fed to broilers there was no effect on meat color and cooking loss, but the pH was significantly reduced. In addition, when garlic bulbs or garlic husks were added to broilers, there was no effect on the cooking loss of chicken meat, but it had a significant effect on pH [67]. However, Oluwafemi et al. [68] reported that the addition of a garlic and ginger mixture did not significantly affect the color of chicken meat, and in another study, the addition of onion extract did not affect chicken quality [69]. In our study, unlike other plants of the genus *Allium*, the addition of CCs did not significantly affect the quality of the chicken meat.

Spleen, liver, and bursa of Fabricius are important immune organs of broilers [70,71]. The bursa of Fabricius is an immune tissue responsible for the development of B lymphocytes and the diversity of the antibody repertoire. It is known to enlarge with the spleen upon stress exposure [71]. In addition, the liver is a tissue that plays a central role in producing energy for broilers such as gluconeogenesis in addition to regulating the body’s immunity. Damage to the liver due to stress can lead to energy loss [70]. Therefore, many researchers have investigated the effects of plants and plant extracts with strong antioxidant activity on the productivity and health of poultry [72,73]. Abou-Elkhair et al. [72] have shown the improved productivity and immunity of broilers when pepper, turmeric powder, and/or coriander seeds were added. The addition, thymol and carvacrol extracted from plants also have positive effects on health by improving the body’s antioxidant power and immune response [73]. Broilers fed garlic powder, a plant of the genus *Allium*, also showed significantly lower liver and immune-related organ weights [74]. As the concentration of garlic powder fermented with *Leuconostoc citreum* increased, the weight of the bursa of Fabricius decreased [75]. The addition of FCC showed positive effects on the immunity parameters of poultry birds, leading to lower spleen weights compared to the control. However, systematic analysis is required through additional studies such as immunoglobulin levels, white blood cell and platelet counts, antibody titer, tumor necrosis factor-α (TNF-α), and interleukin (IL) levels following the addition of FCC. 

Analyses of blood variables are important for the objective assessment of livestock health and for accurately diagnosing disease occurrences [76]. Factors that can affect blood components include season, age, rearing applied environment conditions, nutrition, and feed additives [76,77]. In particular, AST, ALT, and LDH are indicators to be analyzed when evaluating the effects of various feed additives on the liver of poultry. If their levels are lacking or decrease, it means that the added substance is not interfering with liver function [78]. In the blood profile results of this study, the CC-treated group showed lower AST levels than the antibiotic-treated group, which is considered to have a beneficial effect on the health of poultry such as the prevention of liver damage, liver inflammation and liver cancer. In addition, when looking at the significantly lower TCHO levels in the FCC group, it can be judged that the ability of CC-derived flavonols to inhibit adipogenesis is improved through fermentation [52]. Similarly, in the study of Ao et al., when fermented garlic was fed to broilers, the TCHO, TG, and LDL-C levels were significantly lower than those of the control group [66]. In another study, broilers fed garlic and/or onion powder had lower TCHO levels than the controls [79]. The addition of plants of the genus *Allium* is judged to be effective in inhibiting adipogenesis in the body [79].

The addition of the CC feed additive into the feed did not significantly affect the length of the small intestine or the cecum, although it did affect the weight of the small intestine (duodenum, jejunum, and ileum). In addition, when the villi and crypt lengths of the jejunum and ileum were investigated, all treatment groups showed similar results. The structure of the small intestine plays an important role in the digestion and absorption of nutrients [80]. It has been reported that characteristics of the small intestine (length and weight) can affect the ability to digest nutrients [81]. In addition, the villi height is an indicator of the function and activation of the intestinal villi. The ratio of the villi height, villi, and crypts indicates the ability of the small intestine to digest and absorb nutrients [82]. Many factors can affect the characteristics of the small intestine, including manipulation of feed or the addition of feed additives [80,81,82]. Mohiti-Asli and Ghanaatparast-Rashti [83] reported that phytonutrients in feed can improve the length and shape of the intestine. Karangiya et al. [84] reported that garlic supplementation (10 g/kg feed) can improve intestinal absorptive surface area (villi height, width, and crypt depth). Garlic extract fermented with *Lactobacillus acidophilus* can improve digestion by improving the length of the small intestine villi. It also has a strong antioxidant effect with a positive effect on gut health [85].

Many studies have reported the distribution of intestinal microbes in poultry after adding phytogenic feed additives. Recently, Soliman [86] reported that botanical substances such as extracts of plants, herbs, spices, or essential oils can improve feed utilization efficiency, gut microbiome patterns, and stimulate the immune response in poultry due to their antibacterial and antioxidant effects. Mohiti-Asli and Ghanaatparast-Rashti [83] reported that phytonutrients in feed can reduce the abundance of *E*. *coli* and pathogenic fungi in the intestines of poultry birds. Sunu et al. [85] found that fermented garlic can significantly reduce the number of *E*. *coli* and total coliform bacteria in the ileal and cecum and can increase the number of LABs known to have beneficial effects on broilers. Kirkpinar et al. [64] reported that the use of garlic oil alone or in combination with oregano can reduce the number of *Clostridium* in the broiler ileum due to the antibacterial effect of garlic oil. In another study, when onions (0.3%) were added to broiler feed, the number of intestinal LABs was increased, whereas that of *E*. *coli* was decreased compared to the antibiotic (virginiamycin) treatment group [44]. Similarly, in the present study, the addition of CCs inhibited the growth of pathogens but increased the growth of LABs compared to the antibiotic-treated group. 

Plants of the genus *Allium*, especially CCs, contain a large amount of physiologically active substances that have positive effects on the body [14,17,41,52,87]. According to the investigation, there are substances such as glycyrol, tryptophan, and tianshic acid in CCs, and the contents of propanoic acid, benzoylmesaconine, kaempferol, isorhamnetin, quercetin, hydroxystearic acid, and saponins were improved by *L*. *plantarum* SK4719 fermentation [17]. These improved organosulfur compounds (OSCs) and polyphenols have been reported to improve health when ingested with antioxidant, anti-inflammatory, antimicrobial, antiglycemic, and anticancer effects [52]. In particular, it was found that CCs exhibit antibacterial activity against gram-positive and gram-negative bacteria due to the high content of organic sulfur compounds, and in particular, the pH is lowered due to the fermentation of lactic acid bacteria which prevents the proliferation of putrefactive organisms [17]. The stronger antimicrobial activity of FCC may also be due to lactic acid and short-chain organic acids released after fermentation [17]. In conclusion, it is shown that the various physiologically active substances contained in CCs, the improved functional substances, and lactic acid production through fermentation of *L*. *plantarum* SK4719 inhibit the growth of pathogens in poultry and improve health.

## 5. Conclusions

In broilers, dietary supplementation with *L*. *plantarum* SK4719 fermented CC juice lowered the spleen weight and the AST and TCHO levels in the blood and had a positive effect on poultry health while suppressing pathogenic gut bacteria. This study suggests that fermented CC juice can be regarded as a phytogenic feed additive for poultry without side effects as an alternative to antibiotics. However, further studies are needed to demonstrate the effects of CC juice on gut microbiota and immune parameters.

## Figures and Tables

**Table 1 animals-12-02742-t001:** Feed compositions and nutritional compositions of starter and grower feed for broilers.

Item	Starter (1–21 Days)	Grower (22–35 Days)
Ingredient composition, %		
Corn	57.20	52.53
Soybean meal	27.09	26.48
Rice bran	-	3.00
Corn gluten meal	3.38	3.00
Distillers dried grains with solubles (DDGS)	3.00	3.00
Tallow	3.00	5.00
Rapeseed meal	2.00	2.00
Wheat bran	-	1.25
Limestone, 37.2% Ca	1.69	1.79
Mono calciumphosphate (MCP)	1.16	0.91
L-Lysine HCl, 78%	0.38	0.19
DL-Methionine, 96%	0.36	0.17
Salt	0.25	0.25
Choline-Cl 50%	0.09	0.04
Mineral mix ^1^	0.15	0.15
Vitamin mix ^2^	0.12	0.11
Phytase optisphos ^3^	0.05	0.05
NaHCO_3_	0.08	0.08
Calculated nutrient composition		
AMEn, kcal/kg	3100	3150
Crude protein, %	20.50	20.00
Crude fat, %	5.84	8.23
Crude fiber, %	2.80	2.92
Ash, %	4.74	4.97
Calcium, %	0.80	0.90
Phosphorus, %	0.62	0.61
Met+Cys, %	1.03	0.83

^1^ Mineral premix provided the following per kg of diet: Fe, 80 mg; Zn, 50 mg; Mn, 60 mg; Co, 0.3 mg; Cu, 10 mg; and Se, 0.2 mg. ^2^ Vitamin premix provided the following per kg of diet: vitamin A, 80,000 IU; vitamin D_3_, 1600 IU; vitamin E, 20 IU; vitamin K_3_, 8 mg; vitamin B_1_, 8 mg; vitamin B_2_, 24 mg; vitamin B_6_, 12 mg; vitamin B_12_, 0.04 mg; pantothenic acid, 40 mg; folic acid, 4 mg; and nicotinic acid, 120 mg. ^3^ Phytase OptiPhos contained the following per kg: phytase, 1000 FTU.

**Table 2 animals-12-02742-t002:** Comparison of productivity according to dietary feed additive.

Item	Treatment ^1^				SEM ^2^	*p*-Value
	NC	PC	NCC	FCC		
7 days						
BW, g ^3^	134	134	134	134	0.021	0.979
Starter period (8–21 days)						
BW, g	657	714	670	697	10.2	0.183
ADFI, g/d ^4^	63.9	64.0	63.7	66.5	0.822	0.621
ADG, g/d ^5^	37.3	41.4	38.3	40.2	0.725	0.183
F:G, g/g ^6^	1.72 ^a^	1.55 ^b^	1.67 ^a^	1.66 ^a^	0.021	0.022
Grower period (22–35 days)						
BW, g	1520	1654	1555	1584	26.0	0.326
ADFI, g/d	127	134	122	127	1.67	0.102
ADG, g/d	61.6	67.1	63.2	63.3	1.31	0.525
F:G, g/g	2.06	1.99	1.96	2.01	0.035	0.780
Overall period (8–35 days)						
Total gain, g	1385	1519	1420	1449	26.0	0.325
ADFI, g/d	97.2	101	95.5	99.0	1.08	0.253
ADG, g/d	49.5	54.3	50.7	51.8	0.928	0.325
F:G, g/g	1.97	1.87	1.89	1.92	0.025	0.530

^1^ NC, basal diet; PC, basal diet + 0.01% enramycin; NCC, basal diet + 3% non-fermented Chinese chives juice; FCC, basal diet + 3% fermented Chinese chives juice. ^2^ Values are presented as mean ± SEM of five replicates (40 chickens per replicate). ^3^ BW, body weight. ^4^ ADFI, average daily feed intake. ^5^ ADG, average daily gain. ^6^ F:G, feed to gain ratio. a–b Means in the same row with different superscripts differ significantly (*p* < 0.05).

**Table 3 animals-12-02742-t003:** Comparison of the quality characteristics of chicken meat according to dietary CC juice feed additive.

Item		Treatment ^1^				SEM ^2^	*p*-Value
		NC	PC	NCC	FCC		
Weight, g/100 g BW							
Breast		6.79	6.88	6.74	6.81	0.073	0.929
Thigh		6.43	6.32	6.60	6.44	0.066	0.526
Cooking loss, %							
Breast		32.0	30.0	30.2	32.1	0.647	0.530
Thigh		31.4	31.2	28.3	31.2	0.639	0.263
Meat quality							
Breast	L*	57.8	54.8	57.3	58.7	0.598	0.118
	a*	2.59	2.65	3.60	1.98	0.252	0.150
	B*	12.2	12.9	11.7	13.2	0.297	0.289
Thigh	L*	55.1	55.5	57.6	55.4	0.394	0.097
	a*	6.30	6.76	6.51	6.80	0.229	0.870
	b*	14.5 ^a^	11.4 ^c^	12.8 ^bc^	14.3 ^ab^	0.331	<0.001
pH							
Breast		5.59	5.64	5.64	5.56	0.020	0.408
Thigh		6.09	6.21	6.11	6.04	0.035	0.367

^1^ NC, basal diet; PC, basal diet + 0.01% enramycin; NCC, basal diet + 3% non-fermented Chinese chives juice; FCC, basal diet + 3% fermented Chinese chives juice. ^2^ Values are presented as mean ± SEM of five replicates (2 chickens per replicate). a–c Means in the same row with different superscripts differ significantly (*p* < 0.05).

**Table 4 animals-12-02742-t004:** Organ characteristics of broilers according to the feeding of CC juice feed additive.

Item	Treatment ^1^				SEM ^2^	*p*-Value
	NC	PC	NCC	FCC		
	g/100 g BW					
Spleen	0.115 ^a^	0.094 ^b^	0.098 ^ab^	0.086 ^b^	0.004	0.012
Liver	2.20	2.26	2.26	2.06	0.032	0.090
Bursa of Fabricius	0.192	0.211	0.177	0.184	0.007	0.343

^1^ NC, basal diet; PC, basal diet + 0.01% enramycin; NCC, basal diet + 3% non-fermented Chinese chives juice; FCC, basal diet + 3% fermented Chinese chives juice. ^2^ Values are presented as mean ± SEM of five replicates (2 chickens per replicate). a–b Means in the same row with different superscripts differ significantly (*p* < 0.05).

**Table 5 animals-12-02742-t005:** Comparison of blood components according to the feeding of CC additives in broilers.

Item	Treatment ^1^				SEM ^2^	*p*-Value
	NC	PC	NCC	FCC		
AST, U/L ^3^	212 ^b^	252 ^a^	222 ^b^	215 ^b^	5.19	0.019
ALT, U/L ^4^	7.00	6.38	6.50	7.13	0.211	0.535
GGT, U/L ^5^	34.6	43.6	38.6	41.4	1.75	0.307
LDH, mg/dL ^6^	6437	5534	6111	5097	196	0.065
ALP, IU/L ^7^	1,5944	1,3493	1,5494	1,5371	913	0.716
TG, mg/dL ^8^	101	94.9	110	88.6	3.51	0.173
TCHO, mg/dL ^9^	104 ^a^	103 ^a^	96.4 ^a^	86.4 ^b^	1.83	<0.001
HDL-C, mg/dL ^10^	89.0 ^a^	93.5 ^a^	86.3 ^ab^	77.8 ^b^	1.94	0.024
HDL-C, %	85.9	90.9	89.7	90.2	1.07	0.370
LDL-C + VLDL-C, mg/dL ^11^	14.5	9.00	10.1	8.63	1.12	0.225
CRE, mg/dL ^12^	0.990	0.874	0.929	0.814	0.03	0.098
UA, mg/dL ^13^	6.14 ^b^	6.73 ^b^	8.48 ^a^	7.01 ^b^	0.27	0.009
BUN, mg/dL ^14^	2.48	2.49	2.59	2.50	0.05	0.872

^1^ NC, basal diet; PC, basal diet + 0.01% enramycin; NCC, basal diet + 3% non-fermented Chinese chives juice; FCC, basal diet + 3% fermented Chinese chives juice. ^2^ Values are presented as mean ± SEM of five replicates (2 chickens per replicate). ^3^ AST, aspartate aminotransferase. ^4^ ALT, alanine aminotransferase. ^5^ GGT, gamma(γ)-glutamyl transferase. ^6^ LDH, lactate dehydrogenase. ^7^ ALP, alkaline phosphatase. ^8^ TG, triglycerides. ^9^ TCHO, total cholesterol. ^10^ HDL-C, high density lipoprotein-cholesterol. ^11^ LDL-C, low-density lipoprotein-cholesterol; VLDL-C, very-low-density lipoprotein-cholesterol. ^12^ CRE, creatinine. ^13^ UA, uric acid. ^14^ BUN, blood urea nitrogen. a–b Means in the same row with different superscripts differ significantly (*p* < 0.05).

**Table 6 animals-12-02742-t006:** Intestinal characteristics of broilers according to the feeding of CC juice feed additive.

Item	Treatment ^1^				SEM ^2^	*p*-Value
	NC	PC	NCC	FCC		
	cm/100 g BW					
Duodenum	2.10	1.78	1.86	1.98	0.045	0.058
Jejunum	5.03 ^a^	4.13 ^c^	4.27 ^bc^	4.83 ^ab^	0.127	0.025
Ileum	5.57	4.55	4.79	4.69	0.188	0.223
Ceca	2.27	2.74	2.18	2.15	0.145	0.463
	g/100 g BW					
Duodenum	0.579 ^b^	0.450 ^b^	0.578 ^b^	0.868 ^a^	0.036	<0.001
Jejunum	0.946 ^a^	0.708 ^b^	0.875 ^a^	0.933 ^a^	0.028	0.005
Ileum	0.923 ^a^	0.696 ^b^	0.880 ^a^	0.905 ^a^	0.030	0.020
Ceca	0.381	0.336	0.315	0.344	0.021	0.741

^1^ NC, basal diet; PC, basal diet + 0.01% enramycin; NCC, basal diet + 3% non-fermented Chinese chives juice; FCC, basal diet + 3% fermented Chinese chives juice. ^2^ Values are presented as mean ± SEM of five replicates (2 chickens per replicate). a–c Means in the same row with different superscripts differ significantly (*p* < 0.05).

**Table 7 animals-12-02742-t007:** Mean height (μm) of the jejunum and the ileum villi, crypt depth, and villi height/crypt depth ratio of broiler chickens fed with Chinese chives juice.

Item	Treatment ^1^				SEM ^2^	*p*-Value
	NC	PC	NCC	FCC		
Jejunum						
Villus height, μm	790	722	1007	961	61.8	0.337
Crypt depth, μm	211	154	200	208	15.0	0.566
Villus height/Crypt depth	3.95	5.22	5.46	5.30	0.348	0.443
Ileum						
Villus height, μm	792	537	716	1033	64.5	0.138
Crypt depth, μm	194 ^a^	107 ^b^	130 ^b^	188 ^a^	13.2	0.013
Villus height/Crypt depth	5.52	6.33	5.70	6.34	0.351	0.826

^1^ NC, basal diet; PC, basal diet + 0.01% enramycin; NCC, basal diet + 3% non-fermented Chinese chives juice; FCC, basal diet + 3% fermented Chinese chives juice. ^2^ Values are presented as mean ± SEM of five replicates (2 chickens per replicate). a–b Means in the same row with different superscripts differ significantly (*p* < 0.05).

**Table 8 animals-12-02742-t008:** Comparison of the number of microorganisms in the intestines of broilers according to the addition of CC juice feed additives.

Item	Treatment ^1^ (log CFU/g)				SEM ^2^	*p*-Value
	NC	PC	NCC	FCC		
Jejunum						
MacConkey	6.51 ^b^	7.36 ^a^	7.45 ^a^	6.16 ^b^	0.151	<0.001
MRS	8.41 ^a^	7.60 ^b^	7.59 ^b^	8.28 ^a^	0.127	0.019
NA	8.05 ^a^	6.87 ^b^	6.60 ^b^	7.98 ^a^	0.161	<0.001
ST	7.03	6.81	6.53	6.78	0.107	0.470
SS	6.21	6.56	5.65	5.62	0.147	0.051
Ileum						
MacConkey	6.47 ^b^	8.26^a^	7.32 ^b^	7.16 ^b^	0.178	0.004
MRS	9.08 ^a^	8.34 ^b^	8.85 ^a^	8.92 ^a^	0.072	<0.001
NA	7.51 ^b^	8.26 ^a^	7.72 ^b^	8.26 ^a^	0.098	0.010
ST	8.28	8.11	8.32	8.40	0.061	0.493
SS	5.42 ^c^	7.78 ^a^	6.13 ^b^	5.75 ^bc^	0.207	<0.001
Ceca						
MacConkey	7.60	7.56	6.90	7.17	0.120	0.111
MRS	9.26 ^a^	8.50 ^b^	9.14 ^a^	9.10 ^a^	0.097	<0.017
NA	7.80 ^ab^	8.16 ^a^	7.47 ^b^	7.70 ^ab^	0.109	0.158
ST	8.99	8.57	8.81	8.72	0.112	0.622
SS	6.84 ^b^	7.22 ^a^	6.44 ^c^	6.40 ^c^	0.088	<0.001

^1^ NC, basal diet; PC, basal diet + 0.01% enramycin; NCC, basal diet + 3% non-fermented Chinese chives juice; FCC, basal diet + 3% fermented Chinese chives juice. ^2^ Values are presented as mean ± SEM of five replicates (2 chickens per replicate). a–c Means in the same row with different superscripts differ significantly (*p* < 0.05).

## Data Availability

The data are available on request from the corresponding author.

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
