# Peer review of "Evaluation of Non-Fermented and Fermented Chinese Chive Juice as an Alternative to Antibiotic Growth Promoters of Broilers"

_animals, 2022, doi:10.3390/ani12202742_

Round 1
Reviewer 1 Report
The content of the paper “Evaluation of Non-Fermented and Fermented Chinese Chive Jice as an Alternative to Antibiotic Growth Promoters of Broilers ” correspond to a paper title. The paper structure is appropriate – specific for this kind of papers and consists of 5 chapters and references. The order and content of chapters were arranged correctly. The project was written carefully and clearly in English according to valid techniques for such kind of papers. The data were summarized and presented in tables very skillfully. The references used in Introduction chapter were also matched properly.
Taking into consideration fact, that large-scale animal production, especially broilers without anitbiotical growth promoters usage still is a real challenge due to environmental stress on animals, the area of undertaken research is very importnant and applicable in animal production. There is a need to search dietary methods of animal diets supplementation with various bilogically acitve substances that have potential to replace the feed antibiotics. One of such kind of alternative (having in mind its properties) could be non-fermentated and fermented Chinese chieve (Allium tuberosum). IThe aim of the study is clear but in my opinion the weight of three organs (spleen, liver, bursa of Fabricious) is not enough ti conclude about the activity of immune system and should be supported by at least blood indicatiors such as e.g. Ig. The experimet was desinged correctly. In the experiment The Authors assigned a total of 800 one- day-old Ross 308 broiler to one of the four treatments: 1 - basal diet – with control, negative diet; 2 – positive, control diet with 0.01% of antibiotics growth promotor (enramycin); 3 – negative xcontrol diets + 3% non-fermented Chinese chieve juice; 4 - negative xcontrol diets + 3% non-fermented Chinese chieve juice.
There is no justification in the paper why Authirs decided to use such doses of CC juice in such solution). Also reader do know nothing about the chemical composition of the NCC and FCC including biologically active substance included in solution.
Why Authors decieded to incubate the fresh juice for 18 h in 30C? What decided about the the ratio with water in filan product mixed with xoncentrate mixutres?
Not clear in also feed value (chemical composition) of concentrate mixutres in verious treatments. In NCC and FCC 30 g of liquid was added to the briolers’ diet. In consequence the chemical composigtion fo all experimental diets are the same? Maybe the difeerences in e.g. feed intake were also affected by moisture content in feeds?
There is no juxtaposition of feed value of briolers; diets with bird requierement for nutrients.
Neither in Results nor in Discussion chapter there is no explanation of mechanism that could affect changes betweeen treatments. The Authors shoud try to explain the resoon of observed differences between experimental groups.
Line 363-366 – This statemetn does not finnd confirmation in results presented I Table 2
Line 368-382. This paragraph looks like written not not for ths paper.
Line 405-409 – ALT and LDH do not find confimrnation in the table 5.
Reviewer 2 Report
Lee et al. investigate the effects of non-fermented and fermented Chinese chive juice on broilers' productivity, meat quality, blood characteristics, intestinal characteristics, and microbiota associated with intestinal characteristics. The study is very interesting as the interest to develop alternatives to antibiotics and growth promoters has increased. I have two main concerns: i) there is no analysis of bioactive substances in fermented and non-fermented "CC"? ii) fermented product contains probiotic strain (L. plantarum), why did the authors include a group to be treated only with this probiotic strain? Was the ferment subjected to heat treatment for stability? What about the bacterial count of L. plantarum count in the ferment?
Minor comments: Line 50: Please change the word "antibacterial" with "antimicrobial"//2- Line 53: I propose citation of this recent reference: https://doi.org/10.51585/gjvr.2021.3.0014 // Line 57: I propose citation of this recent reference: https://doi.org/10.51585/gjvr.2021.3.0018 //Line 58: Chinese chives should be Chinese chives (CC) // Line 2100: please use the abbreviation of Chinese chives // Line 227: Colony forming unit (CFU)// Discussion: Please use the abbreviation of Chinese chives throughout the manuscript
Round 2
Reviewer 2 Report
Thanks to the authors, all comments have been addressed.